# Genome-Wide Identification of the Ginkgo (*Ginkgo biloba* L.) *LBD* Transcription Factor Gene and Characterization of Its Expression

**DOI:** 10.3390/ijms23105474

**Published:** 2022-05-13

**Authors:** Yating Tian, Xin Han, Yanshu Qu, Yifan Zhang, Hao Rong, Kai Wu, Li’an Xu

**Affiliations:** Co-Innovation Center for Sustainable Forestry in Southern China, Key Laboratory of Forest Genetics and Biotechnology of Ministry of Education, Nanjing Forestry University, Nanjing 210037, China; yttian@njfu.edu.cn (Y.T.); hanxin@njfu.edu.cn (X.H.); ysqu@njfu.edu.cn (Y.Q.); njfuzyf@163.com (Y.Z.); ronghao@njfu.edu.cn (H.R.); wukai@njfu.edu.cn (K.W.)

**Keywords:** *Ginkgo biloba*, *LBD* gene family, expression pattern, stress physiology

## Abstract

Lateral organ boundaries domain (LBD) proteins are plant-specific transcription factors involved in various transcriptional regulation processes. We identified a total of 37 *GbLBD* genes in ginkgo, and based on gene structure and phylogenetic analysis, the *GbLBD* gene family was classified into class I (33, with the largest number of Id genes (16)) and class II (4). The ginkgo *LBD* gene was also analyzed regarding its chromosomal distributions, gene duplications, promoters, and introns/exons. In addition, gene expression profiling and real-time quantitative PCR analysis showed that the expression of 14 *GbLBD* genes differed in six different tissues and three developmental stages. The *GbLBD* gene of class II were highly expressed relative to the class I gene in all tissues and developmental stages, while class Id gene were generally at low levels or were not expressed, especially in seed developmental stages. The expression pattern analysis of cold/drought treatment and IAA/ABA hormone treatment showed that abiotic stress treatment could significantly induce the expression of *GbLBD* gene, of which class II genes played a key role in stress treatment. Our study provides a solid foundation for further evolutionary and functional analysis of the ginkgo *LBD* gene family.

## 1. Introduction

In plants, the transcription factor (TF) family is involved in many important biological processes, and the lateral organ boundaries domain (*LBD*) gene family is a plant-specific TF family. Lateral organ boundaries (LOB) is the first identified member in *LBD* family, which is specifically expressed at the base of the lateral organ in *Arabidopsis thaliana*. A search of the *A. thaliana* genome database using the amino acid sequence encoded by the *LOB* gene as a probe identified 42 homologs of LOB [1]. These homologs all contain a highly conserved LOB structural domain of approximately 100 amino acids in their coding products, thus the gene family is called the *LBD* gene family. The LBD protein consists of a relatively conserved N-terminal region and a variable C-terminal region [2]. The N-terminal contains a fully conserved CX2CX6CX3C (C stands for conserved cysteine residues and X stands for unconserved amino acid residues) zinc finger-like motif, a GAS (Gly-Ala-Ser) region in the middle, and a 30-amino acid LX6LX3LX6L leucine zipper-like helical convoluted motif [1,3]. The LX6LX3LX6L leucine zipper-like helical convoluted motif consists of 30 amino acids [3]. CX2CX6CX3C motif is required for DNA binding, and the LX6LX3LX6L is involved in the interaction of LBD proteins with other proteins [4]. Further studies revealed that the variable C-terminal region has an effect that activates or represses transcription, influencing the expression of downstream genes [5].

Since the first *LBD* gene was reported in *A. thaliana* [3], members of the *LBD* family have been found in many other higher plants, such as *Oryza sativa* [6], *Zea mays* [7], *Populus trichocarpa* [8], *Fragaria vesca* [9], and *Pyrus bretschneideri* [10], with varying numbers of *LBD* gene families in different species. Originally, LBD proteins were thought to play key roles in the development of flowers, leaves, and lateral roots [11]. For example, *AtLBD6* regulates leaf formation [3], *AtLBD16*, *AtLBD18*, and *AtLBD29* control lateral root formation [5] in *A. thaliana*. With the progress of functional studies, LBD proteins were also involved in other plant growth and developmental processes, such as vascular differentiation [12], plant regeneration [13], nitrogen metabolism [14], photomorphogenesis [15], and pathogen resistance [16]. For example, *P. trichocarpa PtLBD1* plays an important role in secondary bast growth [17]. *A. thaliana AtLBD15* produces a large number of secondary metabolites during the metabolism of lignin formation [18]. *AtLBD37*, *AtLBD38*, and *AtLBD39* are negative regulators of anthocyanin synthesis and nitrogen metabolism signaling [14]. *AtLBD20* is involved in the jasmonate signaling pathway, which is mediated by the plant disease-resistance response [16]. In addition, *LBD* genes have similar biological functions in different angiosperms; for example, *AtLBD16* homologs can regulate the development of lateral roots in *A. thaliana*, *Z. mays*, and *O. sativa* [5,19,20].

*Ginkgo biloba*, which is native to China, is an enduring plant that originated approximately 280 million years ago and is known as a “living fossil” in the plant world [21]. Ginkgo is a long-lived dioecious gymnosperm and the only member of the ginkgo family. Because of its special evolutionary status, ginkgo is considered an important species for studying the relationship between epiphytes and cryptogams. In addition to being an important ornamental and timber tree, the leaves of ginkgo are rich in flavonoids and terpenoids and are widely used in the treatment of cardiovascular diseases and cancer [22], and the fruits have antioxidant, antibacterial, insecticidal, and other biological activities, which have important economic value. Therefore, performing studies on the growth, development and resistance of ginkgo at the genetic level is very important. Obtaining the complete sequence of the ginkgo genome will provide a better platform to study the growth and development of ginkgo at the whole genome level. In this study, we identified *LBD* family members from the latest ginkgo genome database; analyze their conserved structures, protein properties, and evolutionary relationships; and study their expression in different tissues, developmental stages and responses to stress treatments. The results provide a foundation for further studies on the functions of *LBD* genes and reveal their molecular mechanisms in ginkgo growth and development.

## 2. Results

### 2.1. Composition of the Ginkgo LBD Gene Family and Its Characteristics

Forty-five *LBD* candidate genes were obtained using in ginkgo, of which 8 genes were removed due to missing LOB structural domains. A total of 37 nonredundant and complete *LBD* genes were finally identified in the ginkgo genome for further analysis.

The 37 GbLBD proteins contained 115 (GbLBD06) to 633 (GbLBD35) amino acids with molecular masses ranging from 13049.83 (GbLBD06) to 70905.07 Da (GbLBD35) (Appendix A). In addition, the isoelectric point range varied widely from 4.41 (GbLBD26) to 9.68 (GbLBD35), reflecting the high complexity of the ginkgo LBD proteins. In addition, cell-PLoc subcellular localization predictions showed that all LBD proteins were located in the nucleus.

### 2.2. Phylogenetic Analysis and Multiple Sequence Alignment

Phylogenetic tree was constructed based on maximum likelihood by the amino acid sequences of 37 ginkgo LBD proteins, 43 Arabidopsis LBD proteins, and 57 Populus LBD proteins (Figure 1). The results showed that 137 LBD proteins from three species were phylogenetically classified into two major classes, class I and class II. Class I had 115 members, 37, 45, and 33 for Arabidopsis, Populus, and ginkgo, respectively; class II had 22 members, including 6 AtLBD, 12 PtLBD, and 4 GbLBD proteins. In addition, class I can be further divided into six subclasses, namely, Ia, Ib, Ic, Id, Ie, and If, and class II can also be subdivided into subclasses IIa and IIb.

In addition, we investigated the evolutionary relationships of ginkgo with other plant *LBDs* (Appendix A). The analysis included the number of *LBD* family members of 17 species, including mosses (*Physcomitrella patens*), ferns (*Selaginella moellendorrffii*), monocotyledons in angiosperms (*Z. mays*, and *O. sativa*, and *Phyllostachys edulis*), and dicotyledons in angiosperms (*Camellia sinensis*, *P. bretschneideri*, *Vitis vinifera*, *Eucalyptus grandis* and *Glycine max*). We found that the number of *LBD* family members was significantly higher in all of class I than in class II.

Multiple sequence alignment was performed for 37 GbLBD proteins using ClustalW, and the conserved motif icons of LBD were created on the WebLogo3 website (Appendix A). The results showed that the LBD proteins all have a highly conserved LOB region at the N-terminal end consisting of approximately 100 amino acids (Figure 2), of which 33 GbLBDs (89.19%) belong to class I and 4 GbLBDs (10.81%) belong to class II.

### 2.3. Chromosome Localization and Synteny Analysis

The 36 *GbLBD* genes were unevenly distributed on 12 chromosomes, and the density of genes on each chromosome was also uneven (Figure 3). The genes were named *GbLBD01-GbLBD36* based on the chromosome they were located on and their position on the chromosome. Only one gene (*Gb_08465*) was localized on a scaffold which was still computationally not anchored to any chromosome or to an unanchored scaffold, and this gene was named *GbLBD37*. According to the definition of gene clusters (two or more genes in a 200 kb segment on a chromosome), we identified two gene clusters in ginkgo *LBD*, the *GbLBD11* and *GbLBD12* genes on Chr04 (class Id) and the *GbLBD23* and *GbLBD24* genes on Chr08 (class IIa).

We further investigated the expansion mechanism of the *GbLBD* gene family by genomic covariance analysis (Figure 4). Statistically, 22 gene pairs were associated with fragment replication events, and two gene groups (*GbLBD11*/*GbLBD12* and *GbLBD23*/*GbLBD24*) were generated by tandem replication events. This indicates that the evolution of the ginkgo *LBD* gene family involves both fragment replication and tandem replication. To better understand the selective pressure on *GbLBD* genes during evolution, we calculated the ratio between nonsynonymous and synonymous substitutions (Ka/Ks) for 11 *GbLBD* homologous gene pairs (Appendix A). The Ka/Ks values of the *GbLBD* gene pairs were generally less than 1, indicating that these genes may have undergone strong purifying selection during the evolutionary process. 

### 2.4. Analysis of Gene Structure and Conserved Motifs

To further investigate the evolutionary relationships between *GbLBD*, we constructed a second phylogenetic tree (Figure 5A). Subsequently, We analyzed the exon–intron structures composition of the coding sequences of 37 *LBD* genes (Figure 5B). The number of introns ranged from 0 to 4, but most of the genes (70%) contained 1 intron. It can be seen that *LBD* genes in the same subclass are usually highly conserved in terms of exon–intron structure and gene length; for example, all 15 members of the Id subclass (except *GbLBD10*) have two exons and one intron and similar gene length. However, there are also some subclasses that have significant structural differences among their members, such as the If subclasses *GbLBD03* and *GbLBD18* both have one intron, but *GbLBD05* has four introns.

A total of 15 conserved motifs were identified in the ginkgo *GbLBD*, which were named Motif1-Motif15 (Figure 5C), and the length of the 15 conserved motifs ranged from 11–48 amino acids, with 3–8 motifs per *GbLBD* gene. Among them, Motif2 and Motif3 are widely present in all *GbLBD* genes, and after SMART checks, these two motifs are the basic regions of the LOB structural domain, and Motif3 is the most typical CX2CX6CX3C motif. Motif1 and Motif4 are only present in class I, whereas Motif7 and Motif8 are only present in class II. Notably, some motifs were detected only in specific subfamilies. For example, Motif 9 is only present in subclass Id, and Motif 13 is only present in subclass IIa.

### 2.5. Ginkgo LBD Promoter Cis-Element Analysis

Some DNA sequences do not require the synthesis of proteins or RNA, but have direct regulatory effects on gene expression, called cis-acting elements. These regulatory sequences are located on the same DNA molecule or on the same chromosome as the gene being regulated. In this study, we identified cis-acting elements in the 2000 bp range upstream of the *GbLBD* gene (Figure 6). The results revealed a variety of cis-acting elements upstream of the *GbLBD* gene. These include light regulation, IAA, ABA, MeJA, low temperature, and drought response elements. Among them, a total of more than 400 light response-related elements were found upstream of 37 *GbLBD* genes, containing an average of 12 per gene, suggesting that the expression of ginkgo *LBD* genes may be regulated by light. IAA-responsive elements were predicted upstream of 28 *GbLBD* genes, a result suggests that the function of the *GbLBD* gene may also be regulated by auxin.

### 2.6. Patterns of GbLBD Expression in Different Tissues

The transcriptome data from six tissues and different developmental stages in ginkgo root, stem, leaf (June 2021, August 2021, October 2021), microstrobilus, ovulate strobilus, and seed (June 2021, October 2021, and January 2022) were used to investigate the expression of *LBD* genes in different subclasses of ginkgo and because class IIa and class IIb members were put into one class description due to the small number of class IIa and class IIb members (Figure 7). The analysis revealed that 37 *GbLBD* genes had different expression patterns. Among them, five genes (*GbLBD01*, *GbLBD04*, *GbLBD23*, *GbLBD32*, *GbLBD35*) were expressed in all tissues. All four genes in class II were highly expressed except for individual genes that were expressed at low levels in individual tissue stages, especially *GbLBD35,* which was highly expressed in all tissues and may play a wide range of roles in the growth and development of ginkgo. In contrast, the relative expression of members in the Id subclass was low throughout the tissue, as the vast majority of genes were not expressed at the macrosporophyll and seed development stages. Among them, *GbLBD06* is not expressed in any tissue, and a functional redundancy likely exists among members of the Id subclass; the members may also function in other tissues or at other developmental stages. Further analysis revealed that *GbLBD28* was significantly more highly expressed in ginkgo seeds during the pre-developmental period (June) than in seeds during the fruit ripening period (October) and the completion of physiological post-ripening (January of the following year); *GbLBD08* and *GbLBD02* were not expressed in June seeds and were expressed in October and January seeds in the following year; and these genes may be involved in different processes of seed development. In summary, the transcriptome sequencing results confirm that the *GbLBD* gene plays an important role in several biological processes.

To verify the reliability of the transcriptome data and functional clustering analysis, we randomly selected 14 *GbLBD* genes and performed qRT–PCR experiments, and the experimental results were largely consistent with the transcriptome sequencing results (Figure 8). The 14 *GbLBD* genes exhibit diverse expression patterns in different tissues or developmental stages. For example, four members of class Ia (*GbLBD01*, *GbLBD04*, *GbLBD07,* and *GbLBD32*) are highly expressed in microstrobilus and ovulate strobilus tissues, suggesting their possible role in flower development. Notably, *GbLBD04* and *GbLBD32* were highly expressed only in microstrobilus, suggesting that these genes may be microstrobilus-specific. In addition, a high expression was observed for *GbLBD21* and *GbLBD31* in stems; *GbLBD16* in roots; *GbLBD25* and *GbLBD35* in leaves; and *GbLBD05*, *GbLBD23*, *GbLBD24,* and *GbLBD33* in seeds. The same gene was differentially expressed at different stages of tissue development; for example, *GbLBD05* was preferentially highly expressed in immature seeds in June, whereas *GbLBD23*, *GbLBD24,* and *GbLBD33* were abundantly expressed in mature seeds in August and October. Overall, the qRT–PCR results support the transcriptome sequencing results.

### 2.7. Expression of GbLBD Genes under Different Abiotic Stresses and Different Hormone Treatments

The roots of 3-month-old ginkgo seedlings were treated with cold, drought, ABA, and IAA stresses. The above 14 *GbLBD* genes were selected for qRT-PCR analysis (Figure 9 and Figure 10). The results showed that abiotic stress and hormone treatment induced the expression of several basal *GbLBD* genes. For example, the expression of most *GbLBD* genes increased significantly and then decreased with increasing cold treatment time, and the expression of *GbLBD04* was upregulated up to 35-fold that of the control. The expression of nine genes (*GbLBD04*, *GbLBD07*, *GbLBD16*, *GbLBD20*, *GbLBD25*, *GbLBD31*, *GbLBD32*, *GbLBD33,* and *GbLBD35*) was significantly induced at the highest value at 3 h of induction and then roughly followed a downward trend from 6 h onward. Most of the *GbLBD* genes reached their highest expression at 12 h of drought stress treatment. Compared to cold stress, the induction of *GbLBD* genes was delayed by drought stress, but *GbLBD07*, *GbLBD20,* and *GbLBD25* were also significantly induced at 3 h of drought stress. *GbLBD07*, *GbLBD21,* and *GbLBD25* showed similar trends in downregulated expression in response to ABA treatment, but all other *LBD* genes were upregulated to varying degrees. At 1 h of IAA treatment, only the transcript levels of *GbLBD16*, *GbLBD24,* and *GbLBD25* were immediately and significantly upregulated, and most *GbLBD* genes were slightly upregulated (less than three-fold) in response to IAA induction. In addition, the expression patterns of some genes showed opposite changes under abiotic stress or hormone treatments. For example, *GbLBD24* was significantly upregulated under cold stress treatment but significantly downregulated under drought conditions; similarly, ABA significantly induced the expression of *GbLBD04*, while IAA treatment suppressed its expression.

## 3. Discussion

As plant-specific transcription factors, *LBD* genes encode a conserved LOB structural domain and are involved in a variety of biological processes. Many angiosperm *LBD* genes have been identified and studied, such as *A. thaliana*, *O. sativa*, etc. However, studies on *LBD* in gymnosperms has been limited by the availability of previous genomes. The high-quality ginkgo genome completed at the chromosome level [23] provides an opportunity not only to study the evolutionary characteristics of the *GbLBD* gene family but also to compare gymnosperms and angiosperms. Here, we identified a total of 37 *GbLBD* genes, which is less than the number of *LBD* family members in most angiosperms. For example, *E. grandis* has 55 *LBD* genes, and *P. trichocarpa* has 57 *LBD* genes. *GbLBD* genes can be divided into two categories: class I (89.19%), class II (10.81%) (Figure 2), and our results proved that the number of *LBD* genes was significantly higher in class I than in class II, which is consistent with previous studies [10,24].

According to the phylogenetic analysis, a total of 137 LBD proteins in the three species were phylogenetically classified into eight subclasses (class I:a-If, class II:a- IIb) (Figure 1). Since there are no ginkgo genes in class Ie, it is speculated that this subclass may be associated with only certain functional properties of angiosperms [3,25]. LBD proteins that tend to share the same phylogenetic branch have similar molecular functions [26], and this provides a reference for predicting the function of GbLBD proteins. In this study, 11 homologs of ginkgo *LBD* with Arabidopsis and 12 homologs of ginkgo *LBD* with Populus were identified by phylogenetic analysis and BLASTp bidirectional comparison (Appendix A). These *GbLBD* genes all belong to class I, while no homologs were found for any of the four members of class II. We speculate that in ginkgo, the *GbLBD* genes of class I may appear earlier and be more conserved than those of class II, which appears later and evolves more rapidly. The conserved motif analysis of *GbLBD* showed (Figure 5C) that all class I members except *GbLBD03*, *GbLBD06,* and *GbLBD30* have complete LX6LX3LX6L motifs, and the number and type of motifs in the *GbLBD* family are widely conserved across evolution. This is consistent with previous studies [6,9].

It has been reported that fragmentary and tandem replication of gene duplication are the two main causes of gene family expansion in plants [27,28]. Although the genome of ginkgo (10.61 Gb) is much larger than that of other species, its *LBD* gene family is low, ranging from about 41% to 86% of the number of *LBD* genes in *A. thaliana* (43), *Z. mays* (44), *G. max* (90), *P. trichocarpa* (57), and *P. bretschneideri* (60). Therefore, we performed an intragenomic covariance analysis of *LBD*. There are 24 pairs of *LBD* gene replication sequences in the ginkgo genome, including 22 pairs of fragment replications and only two pairs of tandem replications, which may explain the relatively conserved number of *GbLBD* gene family members [29].

There is growing evidence that *LBD* genes play a crucial role in the process of plant growth and development, and function of the genes can be predicted with the help of gene expression profiles. Our study found that the *GbLBD* gene of class II was expressed in almost all tissues (Figure 7) and was highly expressed relative to that of class I, which was significantly higher than the Id subclass. In contrast, class I genes are mainly expressed in certain specific tissues. This is consistent with the studies of *LBD* gene family in soybean and wheat [30,31]. Interestingly, the Id subclass *GbLBD* encodes a protein length of 115–197 amino acids, while class II *GbLBD* encodes 324–633 amino acids, which is much greater than the length of the Id subclass *GbLBD*-encoded protein. Class II *LBD* gene were not found to encode proteins of significantly greater length than class I in other species. Therefore, we inferred that during the evolution of ginkgo, the Id subclass *GbLBD* was missing some amino acid sequences, which resulted in low expression (Figure 8). In class Ia, four *LBD* genes (*GbLBD01*, *GbLBD04*, *GbLBD07*, and *GbLBD32*) are highly expressed in floral organs. In contrast, *AtLBD36* and *AtLBD6* in Arabidopsis class Ia are also involved in the regulation of floral organ development and have a degree of functional redundancy [32]. Therefore, we speculate that the four *GbLBDs* may also have functional redundancy in the development of floral organs. Further studies revealed that *GbLBD05* was only highly expressed in immature (June) ginkgo seeds but became less expressed in gradually ripened (August and October) seeds. In contrast, the expression of *GbLBD33* increased significantly with seed maturation, suggesting that *GbLBD33* may be involved in the whole stage of fruit ripening, while *GbLBD05* only plays a role in early fruit development. In contrast, members of the Id subclass are not expressed in the ovulate strobilus or throughout the stages of fruit development. The same phenomenon has been found in other species. In *F. vesca* studies, the *FvLBD18* and *FvLBD25A* genes are involved in early fruit development [9]. During *Musa nana* fruit ripening, the expression of *MaLBD1*, *MaLBD2*, and *MaLBD3* participated in the fruit ripening process by transcriptionally activating the expression of *MaEXPs*, which are relevant to cell wall relaxation factors [33].

Previous research shown that *LBD* transcription factors respond positively to hormonal and abiotic stresses [34]. Our study found that abiotic stress treatments induced a stronger degree of upregulation of *GbLBD* genes compared to that of hormone treatments. The 14 *GbLBD* genes responded to cold, drought, IAA, and ABA stresses which showed the greatest up-regulation were *GbLBD23* (35-fold, IIa), *GbLBD33* (18-fold, Ib), *GbLBD16* (2.5-fold, IIa), and *GbLBD16* (6-fold, IIa), and it was similar to the induction phenomenon of the *LBD* gene in *G. max* [30]. In addition, it is also shown that class II genes play a greater role than class I genes in the plant response to stresses. Studies on Arabidopsis have also shown that class II *LBD* genes (*AtLBD37*, *AtLBD38*, *AtLBD39*, and *AtLBD41*) have the greatest response [16] to a variety of pathogens. The same *GbLBD* gene can respond to multiple stress treatments simultaneously. For example, the expression of the *GbLBD31* gene was not only significantly increased under cold and drought stresses but was also significantly induced by ABA and IAA, indicating that it is a pleiotropic regulator and plays an important role in different signaling pathways. *GbLBD31* is an important candidate gene not only for the response to different abiotic stresses but also for the sensitivity of hormonal stress responses. Here, *GbLBD01* and *GbLBD07* were coregulated by cold and drought stress treatments, suggesting that they may be controlled by both signaling pathways. Arabidopsis *AtLBD15*, which is located in the same branch of the phylogenetic tree, is involved in regulating xylem differentiation [18] and produces a large number of secondary metabolites during the metabolism of lignin formation, thus enhancing the response to abiotic stress. Therefore, we speculate that *GbLBD01* and *GbLBD07* may also be involved in the process of xylem differentiation. In summary, the same gene has diverse adaptive functions in response to different abiotic stresses or hormones, which is important for improving the environmental adaptation of ginkgo by regulating some *GbLBD* genes.

## 4. Materials and Methods

### 4.1. Acquisition of Data and Plant Material

Genome-wide protein files and annotation files of protein-coding genes (in gff3 format) for ginkgo were downloaded from the ginkgo Genome Database (http://gigadb.org/dataset/100613, (accessed on 28 June 2021)). The genomic data of *A. thaliana* and 43 Arabidopsis LBD protein sequences were downloaded from the TAIR10 database (http://www.arabidopsis.org/index.jsp, (accessed on 29 June 2021)) [1]. In addition, the genomic data of *P. trichocarpa* and 57 LBD protein data were obtained from the Phytozome plant genome database (https://phytozome.jgi.doe.gov/pz/portal.html, (accessed on 29 June 2021)) [8].

The ginkgo used in this study were grown on the campus of Nanjing Forestry University, China (32°4′43.45″ N, 118°48′52.01″ E). Tissue samples from Microstrobilus, Ovulate strobilus, June, August, and October leaves, and June, August, and October seed development stages were obtained from seeds that had been postmatured, sprouted hydroponically, and moved to soil for 90 days. Root and stem tissue samples were also obtained. Ginkgo seedlings with a similar growth for 90 days were transferred to low temperature (4°) and drought (15% polyethylene glycol 6000, PEG4000) environments, and leaves were collected at 0, 3, 6, 12, 24, and 48 h; 0.1 mmol/L indole acetic acid (IAA) and 0.1 mmol/L abscisic acid (ABA) concentrations were applied. Young leaves were harvested at 0, 1, 3, 6, 12, and 24 h after treatment. All sampling and treatments were performed in three biological replicates, and the above materials were immediately snap-frozen in liquid nitrogen and stored in a −80 °C refrigerator.

### 4.2. Identification and Distribution Characteristics of Ginkgo LBD Genes

Fasta and Stockholm format files for hidden Markov models of the *LBD* transcription factor family were downloaded from the PFAM website (http://pfam.xfam.org/, (accessed on 30 June 2021)) using the keywords “DUF260 and PF03195”. The HMMER program was used to search for possible *LBD* genes in the full protein file of ginkgo with parameters set to default. In addition, the Fasta format file was compared with the whole protein sequence file for ginkgo using BLASTp (E-value =1 × 10^−^³) to obtain the potential *LBD* genes. The redundant sequences between the two results were removed to obtain the possible *LBD* gene family members. All candidate LBD protein sequences were submitted to the SMART website (http://smart.embl-heideberg.de/, (accessed on 3 July 2021)) and PFAM database for manual correction to obtain eligible *LBD* gene family members. In addition, the ExPASy website (https://web.expasy.org/protparam/, (accessed on 13 July 2021)) was used to predict the physicochemical parameters of all identified LBD proteins. Subcellular localization prediction was performed using the Cell-PLoc 2.0 website (http://www.csbio.sjtu.edu.cn/bioinf/Cell-PLoc-2/, (accessed on 15 July 2021)) [9,10,35].

### 4.3. Multiple Sequence Alignment and Evolutionary Analysis

The core conserved structural domains of these ginkgo *LBD* genes were searched sequentially using the online SMART and PFAM databases, followed by multiple sequence alignment of the core conserved structural domains using ClustalX software [36]; then, the amino acid sequences of the conserved domains were compared and edited using the Jalview software (http://www. jalview.org/, (accessed on 23 August 2021)), which was used to compare and edit the amino acid sequences of the conserved domains. Conserved motif logos were generated using the WebLogo program (http://weblogo.threeplusone.com, (accessed on 23 July 2021)) [37]. Multiple sequence alignment of all LBD protein sequences of ginkgo, *A. thaliana* and *P. trichocarpa* was performed using ClustalW [38], and based on the alignment results, maximum likelihood trees were constructed using MEGA X software with a bootstrap value set to 1000 [39], and the online website Evolview (https://www.evolgenius.info/evolview/, (accessed on 20 July 2021)) was used for visualization [40]. The above method was used to build a maximum likelihood tree based on the identified ginkgo LBD protein sequences.

### 4.4. Synteny Analysis and Ka/Ks Ratio

Information on the distribution of the identified *LBD* genes on chromosomes was extracted from the ginkgo genome database using a native Python script, and chromosome localization was mapped using MG2C (http://mg2c.iask.in/mg2c_v2.0/, (accessed on 20 October 2021)) online software. The CDSs were compared with themselves using BLAST software. Ginkgo LBD family gene duplication events and covariance were detected using MCScanX [41], and the results were visualized using the Advanced Circos function in TBtools [42].

In the Ka/Ks analysis, BLASTp comparison (E-value = 1 × 10^−20^) was used to obtain ginkgo LBD homologous gene pairs, and the set metrics were 1) for long genes, the percentage of regions used for comparison ≥65% and 2) the similarity of regions used for comparison ≥65% [35]. The synonymous substitution rate (Ks), nonsynonymous substitution rate (Ka), and Ka/Ks ratio of homologous gene pairs were calculated using KaKs_Calculator2.0 [34]. The evolutionary divergence time of the *LBD* gene family was calculated using the divergence time formula T= Ks/(2 × 6.1 × 10^−9^) × 10^−6^ Mya [43]. According to the above method, the homologous genes of *GbLBD* with *A. thaliana* and *P. trichocarpa* were obtained based on the phylogenetic tree and BLASTp program bidirectional comparison [35].

### 4.5. Analysis of Ginkgo LBD Gene Structure and Conserved Motifs

The annotation information of the *LBD* gene was extracted from the ginkgo genome annotation file using a Perl language program and was submitted to the Gene Structure Display Server (GSDS: http://gsds.gao-lab.org/, (accessed on 23 July 2021)) to identify the exon–intron structure of the *GbLBD* gene [44]. The online tool MEME (http://meme-suite.org/, (accessed on 24 July 2021)) was used to predict the motifs of LBD proteins with the following parameters: minimum width 6, maximum width 50, number of motifs set to 15, and other parameters were set as default values [45].

### 4.6. Analysis of Cis-Regulatory Elements of the GbLBD Gene Promoter

The cis-acting elements in the 2000 bp promoter region upstream of the transcription start site of each gene were identified using PlantCARE (http://bioinformatics.psb.ugent.be/webtools/plantcare/html/, (accessed on 8 September 2021)), and the results were visualized using TBtools.

### 4.7. Transcription Profiling Based on RNA-Seq Data

In this study, the expression of the *LBD* gene was investigated using transcriptome data from six ginkgo tissues. The transcriptome data of ginkgo leaves (June, August, October) (accession numbers: SRR13517557–SRR13517565) [46] and seeds (August, October, January) (SRR12882901–SRR12882911) [47] were obtained by performing presequencing in our laboratory. Transcriptome data for plant tissues root, stem, Microstrobilus, and Ovulate strobilus were obtained from the NCBI website Sequence Read Archive (SRA) (https://www.ncbi.nlm.nih.gov/sra, (accessed on 1 September 2021)) (SRR7226358–SRR7226374) [48]. RNA-Seq reads from each of the six tissues were mapped to *GbLBD* gene sequences, and the number of reads on each *LBD* gene map was calculated using BWA software (mismatch ≤ 2 bp, the rest of parameters were set as default) [49], and the transcripts per kilobase of exon model per million mapped reads (TPM) values were normalized to Log10 [50].

### 4.8. RNA Isolation and Quantitative Real-Time PCR Analysis

The total RNA of the above samples from different tissues and at different treatment time points was extracted using the Total Plant RNA Extraction Kit provided by Tiangen Biochemical Technology Co., Ltd. Addition of RNase-Free DNase I solution resulted in the removal of DNA effectively. Reverse transcription of RNA to cDNA using PrimeScript^TM^ RT Master Mix (Bao Biological Engineering Co., Ltd., Dalian, China). Real-time quantitative PCR of 14 *GbLBD* genes (Appendix A) was performed on the ViiA7 platform using the PowerUP^TM^ SYBR^TM^ Green Master Mix (Thermo Fisher Scientific, Waltham, MA, USA) fluorescent quantification dye with cDNA as a template. Three technical replicates were set up for each template. The relative expression analysis of genes in different tissues or treatments was performed using the 2^−ΔΔCt^ algorithm with the *Gb_GAPDH* gene as an internal reference gene [51].

## 5. Conclusions

We performed the first systematic identification and analysis of *GbLBD* genes by bioinformatics methods based on ginkgo genome and RNA-seq data, and the 37 *GbLBD* genes identified could be classified into class I (33, with the largest number of Id genes (16)) and class II (4). The gene sequences varied widely in size (from 115–633 amino acids). Class II genes were significantly longer than class I genes. The number and size of genes in the two classes differed significantly. Transcriptomic data from different ginkgo tissues showed that the *GbLBD* gene of class II were highly expressed relative to the class I gene in all tissues and developmental stages, while class Id genes were generally expressed at low levels or were not expressed, especially in seed developmental stages. In addition, 14 *GbLBD* genes were actively involved in ginkgo abiotic stress tolerance and hormone response mechanisms. Abiotic stress treatments induced *GbLBD* genes more strongly compared with the expression pattern of hormone treatments, and the role of class II genes in response to stress treatments was more important than class I genes. Overall, the bioinformatics and expression analysis of *GbLBD* provide a basis for understanding the origin and evolutionary history of ginkgo *LBD*, as well as a reference for further studies on the function of *GbLBD*.

## Figures and Tables

**Figure 1 ijms-23-05474-f001:**
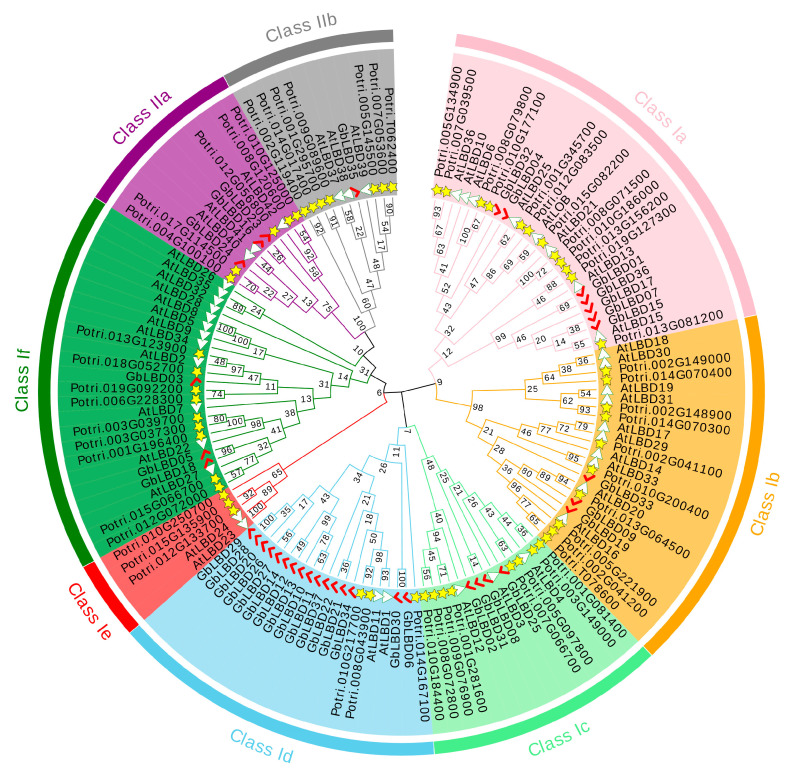
Phylogenetic tree of Lateral organ boundaries domain (LBD) proteins from Ginkgo, Populus, and Arabidopsis. There are eight subclasses in classes Ia-If, IIa and IIb. The red markers, yellow stars and white triangles indicate the LBD proteins in Ginkgo, Populus, and Arabidopsis, respectively.

**Figure 2 ijms-23-05474-f002:**
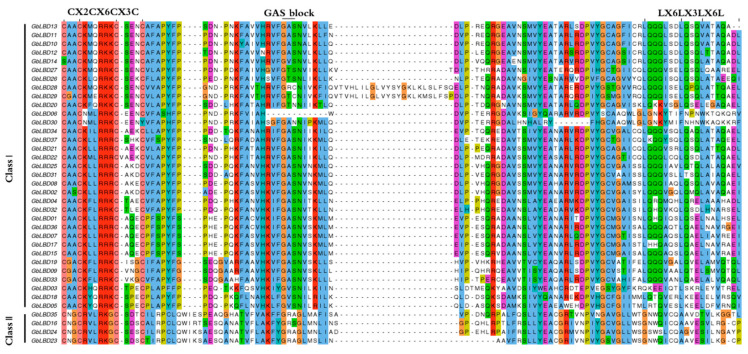
Conserved structural domains of ginkgo GbLBD protein.

**Figure 3 ijms-23-05474-f003:**
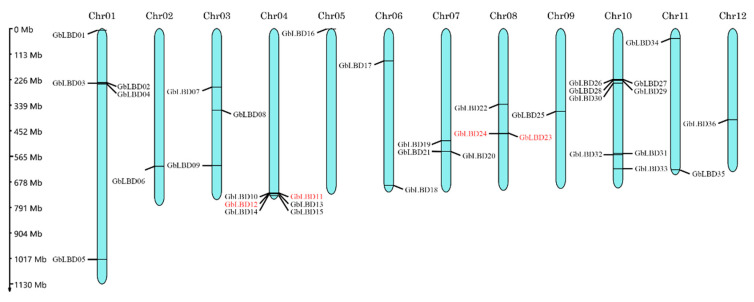
Chromosomal localization of 36 *GbLBD*, chromosome number at the top of each chromosome. Four genes in red font indicate the two gene clusters formed. The scale was in MegaBase (Mb).

**Figure 4 ijms-23-05474-f004:**
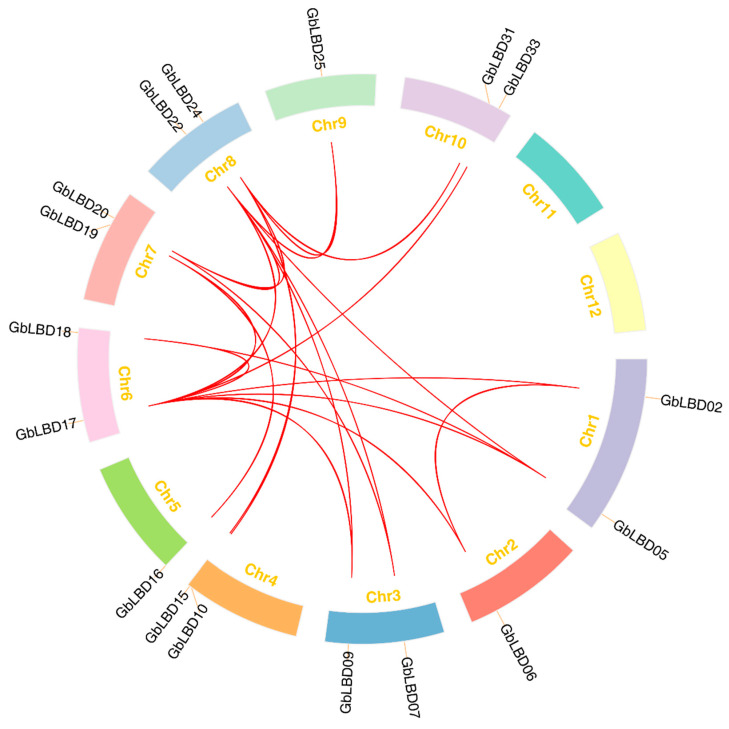
Distribution and synteny of *GbLBD* gene family in ginkgo genome. The red line connects repeated pairs of *GbLBD* genes.

**Figure 5 ijms-23-05474-f005:**
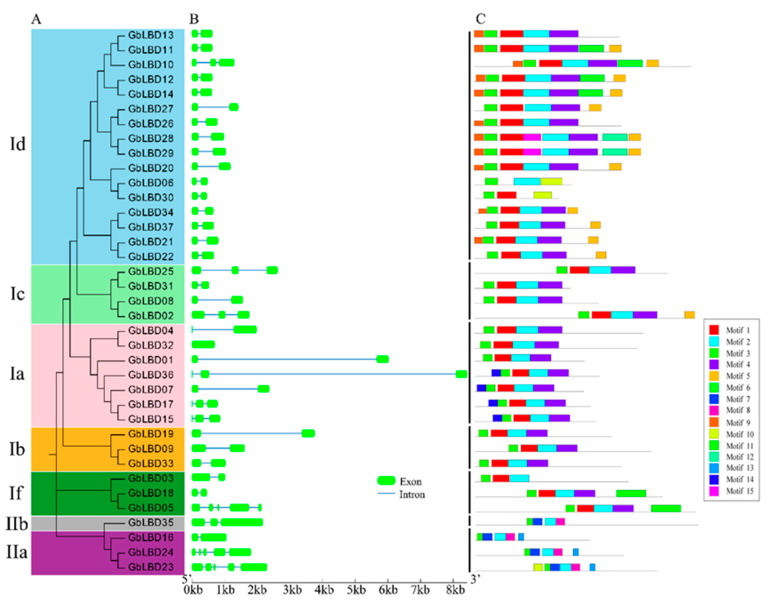
(**A**) Maximum likelihood tree for the GbLBD protein family. Different colors represent different subclasses. (**B**) Exon–intron structure of GbLBD genes. Scale bar indicates 1 kb. (**C**) Distribution of 15 motifs in 37 GbLBD proteins.

**Figure 6 ijms-23-05474-f006:**
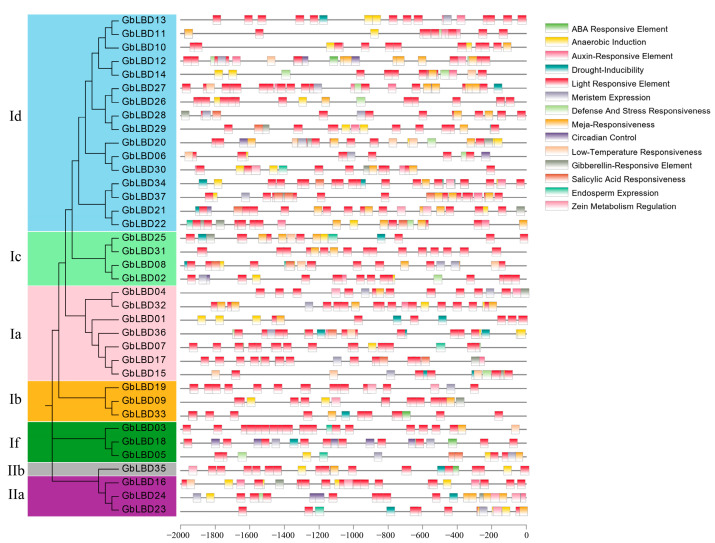
The cis-acting elements in the 2000 bp promoter region upstream of the *GbLBD* gene. The different colors of the rectangular boxes represent different cis-acting elements, and there are elements that overlap each other.

**Figure 7 ijms-23-05474-f007:**
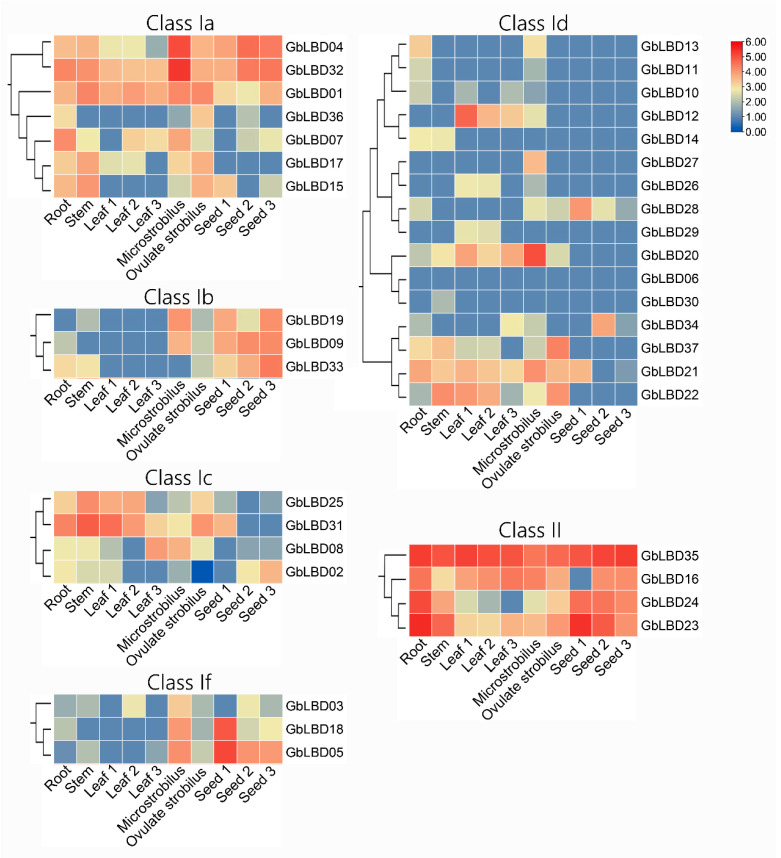
Heatmap of the expression of 37 *GbLBD* genes in six different tissues from the root, stem, leaf, microstrobilus, ovulate strobilus, and seed of plants. Red indicates high expression, blue indicates low expression, and the color scale in the upper right corner indicates the TPM value normalized by log10. Note: Leaf1, Leaf2 and Leaf3 represent June, August and October leaves, respectively; seed1, seed2 and seed3 represent June, October and January seeds, respectively. TPM, Transcripts Per Kilobase of exon model per Million mapped reads.

**Figure 8 ijms-23-05474-f008:**
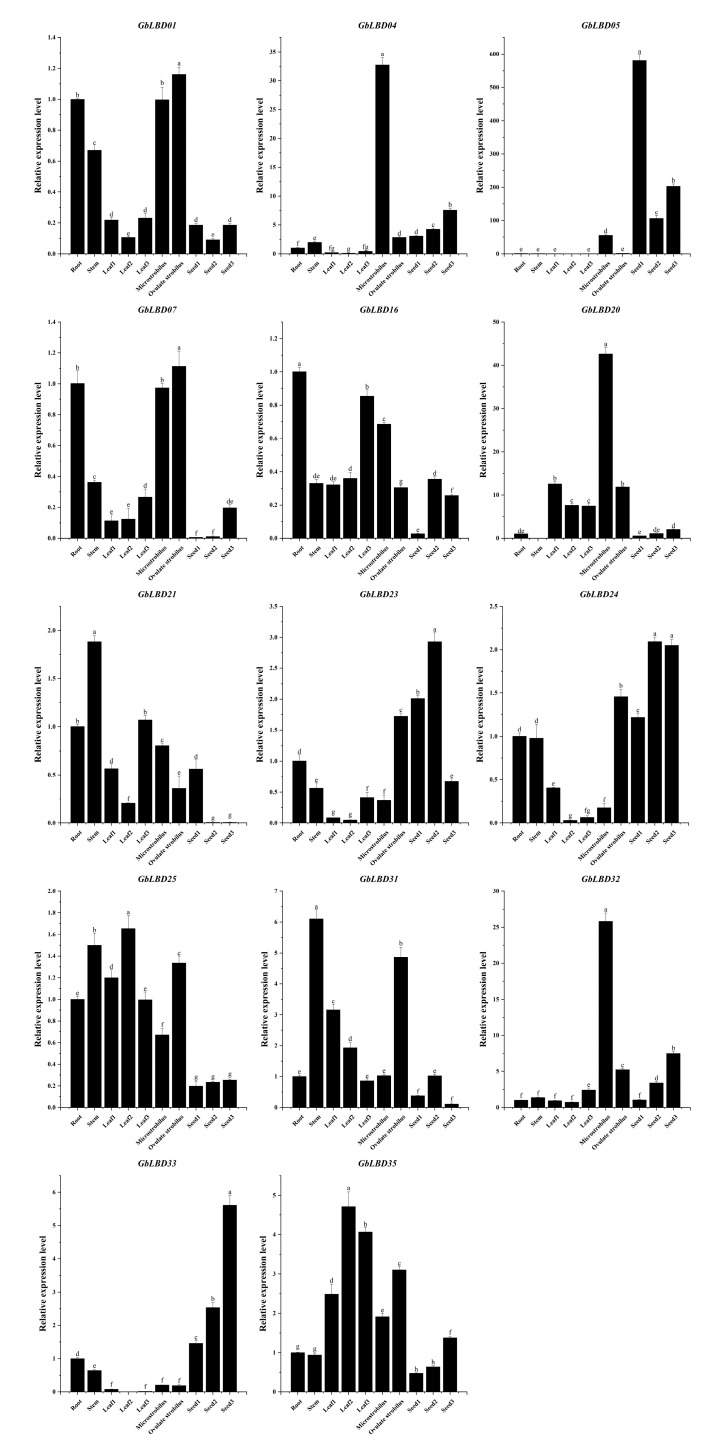
Expression patterns of fourteen *GbLBD* genes in different tissues. The x-axes represent different tissues, including roots, stems, leaves, microstrobilus, ovulate strobilus, seeds; the y-axes indicate the relative expression of *GbLBD* genes. Different letters in a, b, c, d, e, f, g, h indicate significant differences at *p* < 0.05, as determined by one-way ANOVA with Duncan’s multiple range tests. Note: Leaf1, Leaf2 and Leaf3 represent June, August and October leaves, respectively; seed1, seed2 and seed3 represent June, October and January seeds, respectively.

**Figure 9 ijms-23-05474-f009:**
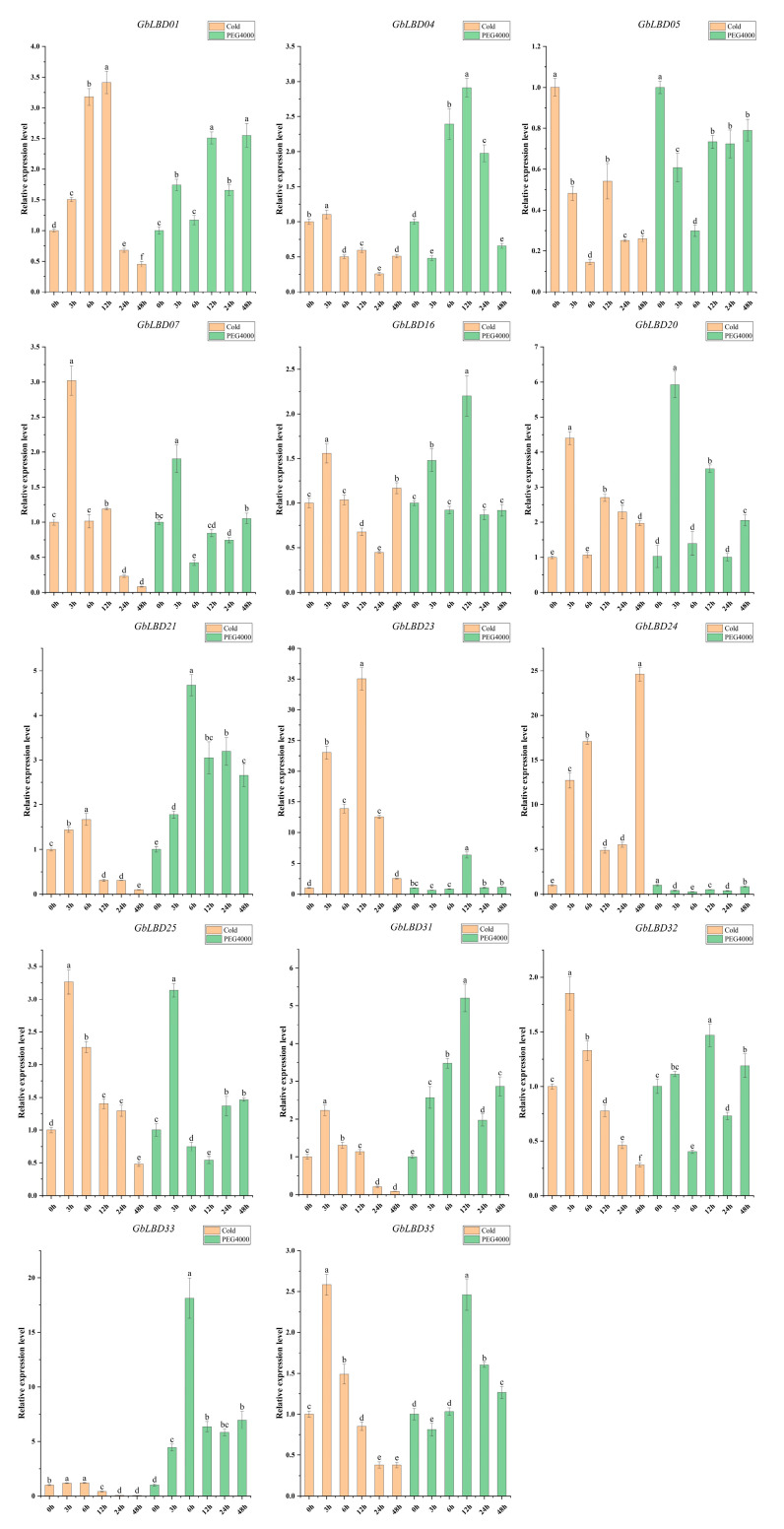
Changes in expression of 14 *GbLBD* genes in response to cold and drought treatments. Different letters in a, b, c, d, e, f indicate significant differences at *p* < 0.05, as determined by one-way ANOVA with Duncan’s multiple range tests.

**Figure 10 ijms-23-05474-f010:**
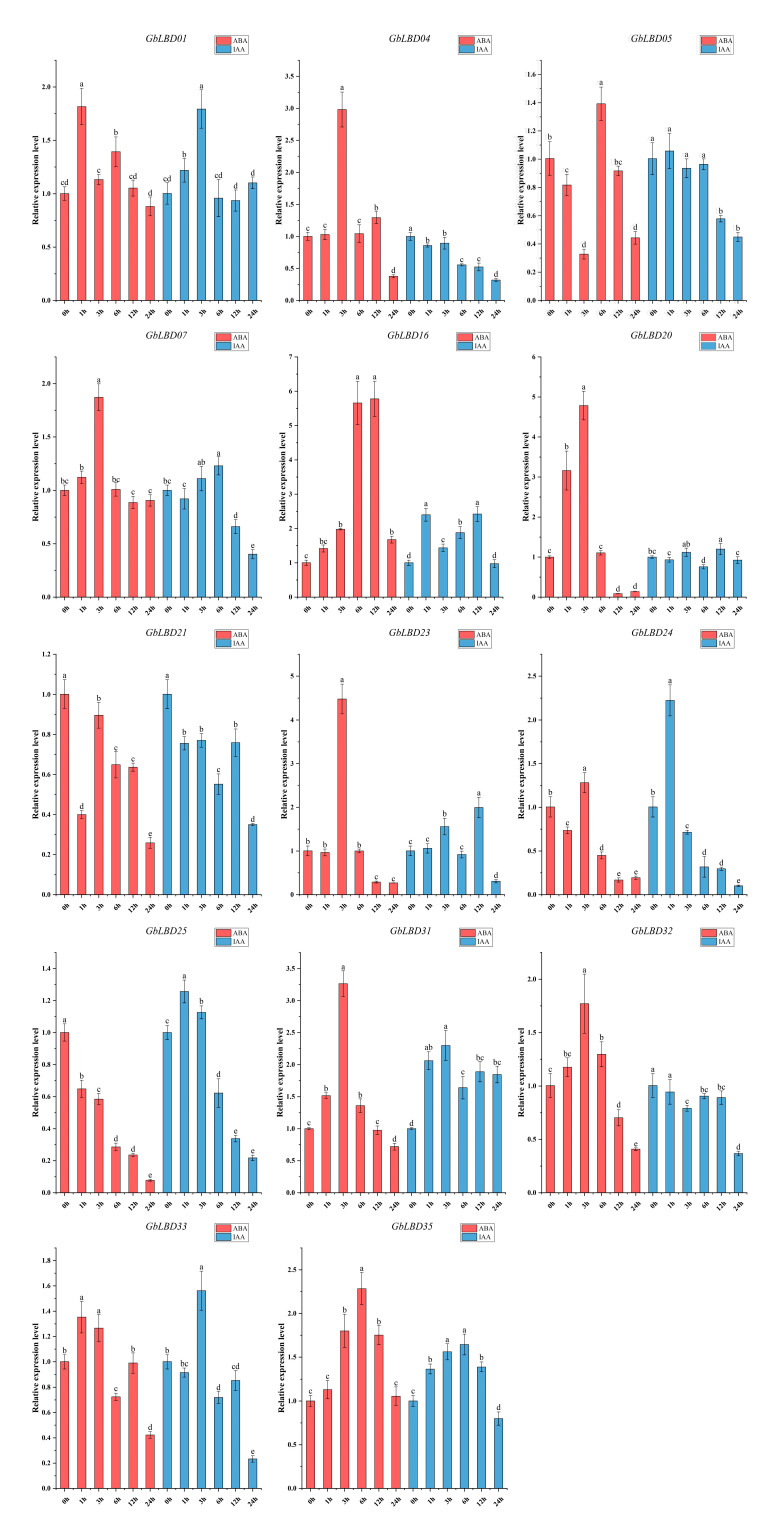
Changes in the expression of 14 *GbLBD* genes in response to ABA and IAA treatments. Different letters in a, b, c, d, e indicate significant differences at *p* < 0.05, as determined by one-way ANOVA with Duncan’s multiple range tests.

## Data Availability

Not applicable.

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
