# Peer review of "Genome-Wide Identification of the Ginkgo (Ginkgo biloba L.) LBD Transcription Factor Gene and Characterization of Its Expression"

_ijms, 2022, doi:10.3390/ijms23105474_

Round 1
Reviewer 1 Report
The manuscript by Tian et al. is a very interesting and important study that clearly demonstrate the need for further analysis LBD family members regarding the ginkgo. Overall, I find the structure of the entire study and the presentation of the methodology valuable. In particular, the main findings that have been supported by PCR- experiments have been adequately presented.
Although I really appreciate the authors' work, the manuscript in its current state still has some problems, which are described in detail below.
In abstract, the authors should write the full name of LBD before using its abbreviation. Further, the sentences are often considerable short and disconnected (e.g. sentences in lines 12,& 13). Please use connecting/linking words between the sentences and avoid too short sentences in the revised version of the manuscript.
Throughout the manuscript, the authors should control the writing style of “class”. It is sometimes written with “C” sometimes with “c”
In line 29: indentified member -> spelling error
In lines 40 to 42: .. Of these CX2… is thought to be… -> The sentence is an interpretation of the authors. But in the introduction, they should not interpret the findings of a previous study. Please re-write this sentence
In line 47: … (Chen et al. 2018) -> It is either a citation error or I could not understand what the authors aim with this citation.
In the complete manuscript, the authors uses the LBD genes as well as proteins interchangeable. Sometimes they started a sentence with LBD genes but make explanation about proteins or vice versa. This situation makes whole manuscript too difficult for the readers. I highly recommend that the authors should find a way to clearly separate the genes from proteins (e.g. lines 96 to 100).
The sentence in lines 173 to 175 is difficult to understand. Especially, the meaning of “flanking sequences” is not introduced. Further, the meaning of “”exercising functional genes” is not clear for me. When is a gene functional and when not?
In line 176: in the 2000 bp range upstream of the GblBD gene: The reason is missing. The region of 2000 bp is huge for a gene promoter, or it could be overlapped with other genes. From my own studies, I know, there are often overlapping between a promoter of a gene and the introns/exons of another gene if we consider such a huge promoter. The authors should ensure that there is not an overlap and give a reason why they consider these promoter regions of 2000 bp but not 1000 or 500 or even shorter?
In lines 178 to 182: did authors perform an enrichment analysis if yes which method. If not, how they interpret the functional importance of these elements.
In line 198: the gene Gblbd35 which is found for all tissues. The authors hypothesis that this gene may play an important role. But per definition, it could be a housekeeping gene as well. The authors should address this point, as well.
In line 474: … this paper provide a valuable reference. The sentence could be interpreted as an overestimation of general findings in this manuscript. Please re-write it.
Author Response
Dear reviewer,
Thank you very much for your positive and constructive comments and suggestions on our manuscript. These comments are very helpful to us in revising and improving the paper, and they are also important guidance for our research. We have paid attention to the comments, and revised the manuscript in the hope that it will be approved. Our point-by-point response to the comments can be found below.
- The manuscript by Tian et al. is a very interesting and important study that clearly demonstrate the need for further analysis LBD family members regarding the ginkgo. Overall, I find the structure of the entire study and the presentation of the methodology valuable. In particular, the main findings that have been supported by PCR- experiments have been adequately presented. Although I really appreciate the authors' work, the manuscript in its current state still has some problems, which are described in detail below.
Response: Thank you very much for your approval of the manuscript and for your very valuable suggestions. We will certainly revise this manuscript carefully and look forward to your approval. Thank you again for your comments.
- In abstract, the authors should write the full name of LBD before using its abbreviation. Further, the sentences are often considerable short and disconnected (e.g. sentences in lines 12, & 13).Please use connecting/linking words between the sentences and avoid too short sentences in the revised version of the manuscript.
Response: Thank you very much for your advice. We have corrected the Lateral Organ Boundaries Domain (LBD) in the revised version; and made changes to avoid incoherent sentences (Line12-14, Line16-19).
- Throughout the manuscript, the authors should control the writing style of “class”. It is sometimes written with “C” sometimes with “c”.
Response: Thank you for pointing this out. In the revised manuscript, we change “class” to “c” in all cases.
- In line 29: indentified member -> spelling error.
Response: Thank you for pointing this out. We apologize for our negligence and have corrected it in the text to “identified member” (Line30).
- In lines 40 to 42: . . Of these CX2… is thought to be… -> The sentence is an interpretation of the authors. But in the introduction, they should not interpret the findings of a previous study. Please re-write this sentence.
Response: Thank you for this suggestion. Thank you for your comments; we have rewritten this sentence in the revised manuscript (Line41-42).
- In line 47: … (Chen et al. 2018) -> It is either a citation error or I could not understand what the authors aim with this citation.
Response: Thank you very much for your advice. We have removed the citation error and are very sorry for our mistake.
- In the complete manuscript, the authors uses the LBD genes as well as proteins interchangeable. Sometimes they started a sentence with LBD genes but make explanation about proteins or vice versa. This situation makes whole manuscript too difficult for the readers. I highly recommend that the authors should find a way to clearly separate the genes from proteins (e.g. lines 96 to 100).
Response: Thank you for this suggestion. We have clearly separated genes and proteins in several places in the revised manuscript (For example, Line89, Line98, Line309-310).
- The sentence in lines 173 to 175 is difficult to understand. Especially, the meaning of “flanking sequences” is not introduced. Further, the meaning of “exercising functional genes” is not clear for me. When is a gene functional and when not?
Response: Thank you very much for your suggestions. We are sorry that we had problems with the interpretation of this section. After discussion, we have made changes in the revised version (Line171-173).
- In line 176: in the 2000 bp range upstream of the GblBD gene: The reason is missing. The region of 2000 bp is huge for a gene promoter, or it could be overlapped with other genes. From my own studies, I know, there are often overlapping between a promoter of a gene and the introns/exons of another gene if we consider such a huge promoter. The authors should ensure that there is not an overlap and give a reason why they consider these promoter regions of 2000 bp but not 1000 or 500 or even shorter?
Response: Thank you for pointing this out. Ginkgo belongs to gymnosperms, with a large genome and redundant information, and a lot of information is unknown. We chose a promoter region of 2000 bp, which is a more mainstream choice(Xie et al.,2020; Song et al.,2020), in order not to miss some information. Also we verified that there was no overlap between genes.
â‘ Xie T, Zeng L, Chen X, et al. Genome-Wide Analysis of the Lateral Organ Boundaries Domain Gene Family in Brassica Napus [J]. Genes (Basel), 2020, 11(3).
â‘¡Song, B. B.; Tang, Z. K.; Li, X. L.; Li, J. M.; Zhang, M. Y.; Zhao, K. J.; Liu, H. N.; Zhang, S. L.; Wu, J., Mining and evolution analysis of lateral organ boundaries domain (LBD) genes in Chinese white pear (Pyrus bretschneideri). BMC GENOMICS 2020, 21, (1).
- In lines 178 to 182: did authors perform an enrichment analysis if yes which method. If not, how they interpret the functional importance of these elements.
Response: Thank you very much for your comments; we apologize that we did not perform an enrichment analysis and have revised the manuscript for any inappropriate descriptions (Line180-181).
- In line 198: the gene Gblbd35 which is found for all tissues. The authors hypothesis that this gene may play an important role. But per definition, it could be a housekeeping gene as well. The authors should address this point, as well.
Response: Thank you very much for your advice. We emphasize that the high expression of the GbLBD35 gene is relative to other genes, but the expression of GbLBD35 in different timespaces belongs to a differential expression pattern, which is inconsistent with the definition that housekeeping genes can be stably expressed in all tissues. Therefore, we did not consider the GbLBD35 gene as a housekeeping gene in our study.
- In line 474: … this paper provide a valuable reference. The sentence could be interpreted as an overestimation of general findings in this manuscript. Please re-write it.
Response: Thank you very much for your recommendation. We have rewritten this sentence in the revised manuscript (Line472-475).
Reviewer 2 Report
Tiang et al have performed the genome-wide identification and characterization of the LBD transcription factor gene family in Ginkgo biloba. Authors have also performed the evolutionary analysis and classified the gene family into several sub-classes. The expression analysis performed during the several vegetative and reproductive stages of the life cycle and in response to several stress conditions clearly shows the importance of the LBD gene family.
The study is very comprehensive and the manuscript is very well written. Following are certain suggestions for further improvement of the manuscript:
- Section 2.3 line 127: As authors knows, genes are always localized on the chromosomes. The statement in line 127 gives an impression that one gene is not present on chromosome. Authors could improve their statement by saying that one gene is localized on a scaffold which is still computationally not anchored to any chromosomes or to an unanchored scaffold.
- Authors should change the "chromosomal distribution.... of all GbLBD genes" in the figure 4 figure legend to some other appropriate word. This data does not represent all genes and also its primarily not intended to show chromosomal distribution.
- Almost 70% of the GbLBD genes are with only 1 intron. Was this the same case for LBD genes in other species?
- Figure 6, x-axis of figure showing cis-element distribution should have the orientation marked with minus (-) for eg: -10 bp to represent the upstream position.
- Figure 8, please change the style of the representation for better visibility. more simpler bar chart could be better option.
- In discussion, line 299: Though the statement about the gene number comparision (which are 0.86, 0.84, 0.41, 0.64 and 0.62 times greater than those of) is technically correct but this give a wrong idea that the gene number in Gb is higher. Authors should simplify the statement.
- Please include a statement about DNase treatment in the qPCR method section. Also, correct the "PowerUPTM SYBRTM Green Master Mix". Either superscript the "TM" which represents trademark or remove them.
Author Response
Dear reviewer,
Thank you very much for your positive and constructive comments and suggestions on our manuscript. These comments are very helpful to us in revising and improving the paper, and they are also important guidance for our research. We have listened carefully to the comments, revised the manuscript, and hope it will be approved. Our point-by-point response to the comments can be found below.
- Section 2.3 line 127: As authors knows, genes are always localized on the chromosomes. The statement in line 127 gives an impression that one gene is not present on chromosome. Authors could improve their statement by saying that one gene is localized on a scaffold which is still computationally not anchored to any chromosomes or to an unanchored scaffold.
Response: Thank you for this suggestion. We have revised the related parts (Line124- 126).
- Authors should change the "chromosomal distribution.... of all GbLBD genes" in the figure 4 figure legend to some other appropriate word. This data does not represent all genes and also its primarily not intended to show chromosomal distribution.
Response: Thank you very much for your suggestion; we have corrected the legend of Figure 4 in the revised manuscript.
- Almost 70% of the GbLBD genes are with only 1 intron. Was this the same case for LBD genes in other species?
Response: Thank you very much for your question. 70% of ginkgo LBD genes contained only 1 intron, which was similar to LBD genes of other species.For example, 70% of the LBD genes in white pear also contain only 1 intron(Song et al.,2020); 69% of the LBD in moso bamboo(Huang et al.,2021); 72% of the LBD in soybean(Yang et al.,2017), etc. Since this is a more common phenomenon in the LBD gene family, we did not focus on it in the paper.
â‘ Song, B. B.; Tang, Z. K.; Li, X. L.; Li, J. M.; Zhang, M. Y.; Zhao, K. J.; Liu, H. N.; Zhang, S. L.; Wu, J., Mining and evolution analysis of lateral organ boundaries domain (LBD) genes in Chinese white pear (Pyrus bretschneideri). BMC GENOMICS 2020, 21, (1).
â‘¡Huang, B.; Huang, Z. N.; Ma, R. F.; Ramakrishnan, M.; Chen, J. L.; Zhang, Z. J.; Yrjala, K., Genome-wide identification and expression analysis of LBD transcription factor genes in Moso bamboo (Phyllostachys edulis). BMC PLANT BIOLOGY 2021, 21, (1).
â‘¢Yang, H.; Shi, G. X.; Du, H. Y.; Wang, H.; Zhang, Z. Z.; Hu, D. Z.; Wang, J.; Huang, F.; Yu, D. Y., Genome-Wide Analysis of Soybean LATERAL ORGAN BOUNDARIES Domain-Containing Genes: A Functional Investigation of GmLBD12. PLANT GENOME 2017, 10, (1).
- Figure 6, x-axis of figure showing cis-element distribution should have the orientation marked with minus (-) for eg: -10 bp to represent the upstream position.
Response: Thank you very much for your advice. we have marked the orientation of the x-axis in Figure 6 with a minus sign (-) in the revised manuscript.
- Figure 8, please change the style of the representation for better visibility. more simpler bar chart could be better option.
Response: Thanks for your comments. We have changed Figure 8 to a simpler bar chart in the revision manuscript.
- In discussion, line 299: Though the statement about the gene number comparision (which are 0.86, 0.84, 0.41, 0.64 and 0.62 times greater than those of) is technically correct but this give a wrong idea that the gene number in Gb is higher. Authors should simplify the statement.
Response: Thanks for your comments. We have simplified the statement in the revision (Line295-298).
- Please include a statement about DNase treatment in the qPCR method section. Also, correct the "PowerUPTM SYBRTM Green Master Mix". Either superscript the "TM" which represents trademark or remove them.
Response: Thanks for your kind suggestion. We include a statement about DNase treatment in the qPCR method section (Line448-449). And, we superscripted "TM" which represents the trademark.
Round 2
Reviewer 1 Report
The authors have considered and addressed all of my comment very intensively. Thank you for that.
From my point of view, the manuscript is now really nice and acceptable.
Congratulations!